# Effect of Diets and Low Temperature Storage on Adult Performance and Immature Development of *Phyllonorycter ringoniella* in Laboratory

**DOI:** 10.3390/insects10110387

**Published:** 2019-11-04

**Authors:** Shubao Geng, Heli Hou, Chuleui Jung

**Affiliations:** 1School of Agricultural Sciences, Xinyang Agriculture and Forestry University, Xinyang 464000, China; shubao.301@163.com; 2Agricultural Science and Technology Research Institute, Andong National University, Andong 36729, Korea; 3School of Food Sciences, Xinyang Agriculture and Forestry University, Xinyang 464000, China; helihou@163.com; 4Department of Plant Medicals, Andong National University, Andong 36729, Korea

**Keywords:** *Phyllonorycter ringoniella*, rearing method, oviposition preference, cold storage, body size

## Abstract

The Asiatic apple leaf miner, *Phyllonorycter ringoniella* (Matsumura), is an important insect pest to apples. We developed a rearing method for *P. ringoniella* in the laboratory. Overwintering pupae were collected from our apple orchard, and crabapple seedlings were selected as oviposition substrate and food source for the larval development. The mean developmental period was 25.9 ± 0.49 days from egg to adult and the survival rate was 0.387 under 25 ± 1 °C, 65 ± 5% RH and a photoperiod of 14:10 (L:D) h. Mean egg length and egg width were 0.336 ± 0.0043 and 0.259 ± 0.0046 mm, respectively. Mean body length and head width increased from 1.070 ± 0.0245 and 0.180 ± 0.0021 mm in first instar larvae to 5.027 ± 0.0718 and 0.321 ± 0.0021 mm in fifth instar larvae, respectively. The mean weight of the pupae was 0.946 ± 0.0132 mg. The wingspan of male adult (6.280 ± 0.0639 mm) was significantly larger than female (6.040 ± 0.0753 mm). The mean fecundity (56.9 ± 8.58 eggs per female) and longevity (8.2 ± 0.55 days) of females was highest when they were provided with 10% honey solution, followed by 10% sugar, water, and control. The females preferred to lay eggs (56.5 ± 3.84%) near the secondary leaf vein in a leaf. The larval mortality increased linearly (R^2^ = 0.94) with the number of larvae per leaf. The mortality of pupae increased from 5.6 ± 4.01 to 51.1 ± 9.88% as storage periods at 4 °C increased from 0 to 105 days. The rearing method and its biological characteristics presented in this study will contribute to further studies on this pest insect.

## 1. Introduction

The Asiatic apple leafminer, *Phyllonorycter* (*Lithocolletis*) *ringoniella* (Matsumura) (Lepidoptera: Gracillariidae), is an important insect pest on apple trees with four to six generations per year in Korea, Japan, and China [1,2,3,4,5,6]. The hosts of *P. ringnoniella* include some pome and stone fruit trees such as apple, pear, peach, cherry, and plum [1,7]. Its larva develops in a mine made on the underside of a leaf, and pupates inside of the mine. In an early infestation, silvery-green spots of irregular shape on the lower surface of the leaf and greenish-white appearance on the upper surface can be noticed [8]. The mines caused by *P. ringoniella* can reduce the photosynthetic areas, hasten defoliation, and inhibit the growth of new buds, which may cause premature ripening and fruit drop [3,9,10]. *Phyllonorycter ringoniella* overwinters as pupae inside the mine of defoliated leaves in October to November [11], and emerges from March to April in the following year [2,3,5,10,12,13]. Since 1950, many outbreaks of the pest happened in Japan after application of pesticides [12]. Its pestiferous pressures expanded into the major apple-growing regions in China, and the affected areas were as high as 80% in some outbreak years [7,10].

Only a few studies had dealt with the partial biology of *P. ringoniella* on temperature-dependent development, seasonal occurrences, and its mass flight activity [2,3,4,5,9,13,14,15]. Unfortunately, information on rearing methods of *P. ringoniella*—adult performances supplied with different diets, low temperature storage of pupae, and its morphologic characteristics at different stages, useful for further studies—was limited. Therefore, the purposes of this study were: (1) to develop an ex-situ rearing of *P. ringoniella* using crabapple leaves and to provide morphological characteristics of the different stages *P. ringoniella*; (2) to determine the effects of diets on adult performances; (3) to test the effect of cold storage of early pupa on pupal development and subsequent adult performances; (4) and provide the oviposition pattern on leaves and larval mortality.

## 2. Materials and Methods

### 2.1. Rearing P. ringoniella Using Crabapple Leaves and Morphological Measurements

Overwintering pupae of *P. ringoniella* were originally collected from an apple orchard in Hogye, Mungyeong, Gyeongbuk, South Korea (36°41′ N, 128°13′ E) in 2016. The pupae were separately placed in small Petri dishes (5.5 cm diameter, 1.5 cm height) and then kept in the growth chamber at a temperature of 25 ± 1 °C, 65 ± 5% RH, and a photoperiod of 14:10 (L:D) h. After emergence, 10–20 pairs of the adults were transferred into a plastic cubic breeding cage (40 × 40 × 40 cm, side ventilation) and supplied with 10% sugar solution. The cotton moistened with 10% sugar solution was changed every other day.

Crabapple *Malus prunifolia.* (Rosales: Rosaceae) seedlings were selected as oviposition site and food source of larval development since they are easy to maintain in the laboratory [5]. To assess the larval development, three-month-old crabapple seedlings were pruned to a standard height of 20–30 cm to facilitate handling and to enable the use of standard-sized breeding cage. Each seedling was planted in a single plastic pot (10 cm diameter, 10 cm height) filled with the commercial soil “Plant World”, Korea.

A crabapple seedling was then introduced into the breeding cage for 24 h for the collection of eggs. The eggs on the seedlings were then placed in the growth chamber until they developed into pupae. The newly developed pupae (<24 h) within the leaf mines were separately placed in the small Petri dishes (5.5 cm diameter, 1.5 cm height) for producing the next generations.

In order to evaluate the effectiveness of our rearing method, 150 eggs on 17 seedlings were placed in the growth chamber with temperature at 25 ± 1 °C, 65 ± 5% RH, and a photoperiod of 14:10 (L:D) h. The developments and survivals of eggs were checked daily using a binocular stereomicroscope (Olympus SZ51, Olympus Corporation, Tokyo, Japan). Larval development was monitored continuously from the egg stage onward. Once pupation occurred, pupae were placed individually into petri-dishes (5.5 cm diameter, 1.5 cm height) with a piece of cotton moistened with distilled water to maintain humidity. Pupal development was observed daily until adult emergence or death. The mean developmental periods of eggs, larvae, pupae, and total immature stages were calculated. The survival rate of eggs was calculated as that the number of hatched eggs divided by the number of initial eggs (here is 150), the survival rate of larvae was calculated as the number of larvae that survived to the pupal stage divided by the number of hatched eggs, the survival rate of pupae was calculated as that the number of emerged adults divided by the number of larvae that survived to the pupal stage, and the survival rate of total immature stages was calculated as that the number of emerged adults divided by the number of initial eggs. The sex ratio was determined by the number of female adults divided by total emerged adults.

Additionally, sizes of eggs, larvae, pre-pupae, pupae, and adults of *P. ringoniella* were measured with the calibrated ocular micrometer using a binocular stereomicroscope. The mean length and width of 30 eggs were measured. The larvae from the colony were identified to five instars by the number of head capsules inside the mine. The body lengths and head widths of 30 to 49 larvae at each instar and pre-pupal stage were measured. In addition, the weights of 10 pupae were measured by electrical balance (ALC-310.4, Sartorius Group Brand, Berlin, Germany) serving as one replicate, with a total of 30 replicates being conducted. Body lengths, antenna, and wingspans (fore wing) of female and male adults were also measured as 30 replicates.

### 2.2. Effect of Diets on Adult Performance

Longevity and fecundity of *P. ringoniella* adults were determined when they were supplied with different diets at 25 ± 1 °C, 65 ± 5% RH and a photoperiod of 14:10 (L:D) h. The newly emerged adults (<24 h) were coupled (one female and one male) separately into the plastic transparent cuboid cages (6.5 × 6.5 × 9.5 cm) for copulating and oviposition. The oviposition cage was ventilated from the upper surface and two opposite sides with fabric mesh, and its bottom was hollow. A lid with a small plastic tube (1.5 cm diameter, 6.0 cm height, 5 mL volume) was glued in the center of the cage. The lid could be removed and reinstalled from the cage for experimental processes. One leaf pruned together with its stem from the potted crabapple seedlings was inserted into the plastic tube (filled with tap water) as an oviposition substrate. A piece of cotton moistened with different diet was placed in each cage as food for the adults. The diets were supplied with distilled water, 10% sugar, and 10% honey solutions. In addition, adults neither provided with the oviposition substrate nor the food were regarded as the control group. The number of adults survived (longevity) and the number of eggs laid (including information where on the leaf and the sides of the cage) were recorded daily. Leaves and diets were changed daily. Thirty-six pairs of adults were tested in each group and the data came from egg-laying adult pairs were used in data analyses. The number of adult pairs used in data analyses, were 30, 26, 28, and 24 in distilled water, 10% sugar solution, 10% honey solution, and control group, respectively.

### 2.3. Effect of Low Temperature Storage of Pupae on Adult Performance

Because the eggs and larvae were dependent on leaves being alive, storage of pupae would be the best choice for discontinuous or mass rearing of *P. ringoniella* colonies. In order to test the storage stability of pupae, the developmental period, mortality, and deformation of pupae after different storage periods at 4 °C were examined. The newly developed pupae (<24 h) from the colony were separately placed in the small Petri dishes and transferred to the 4 °C refrigerator. The storage periods of pupae increased from 0 to 105 days by 15 day intervals. The experiment of each storage period included three replications and each replication included 30 pupae. After each storage period, the pupae were taken out from the refrigerator and placed in the growth chamber at 25 ± 1 °C. Development, survival, and deformations were recorded daily. Pupae that could not emerge and turn into adults in 15 days were regarded as dead. The pupae that emerged to adults with malformed wings were considered ‘deformed’. Mortalities and deformation rates of the pupae were calculated by dividing the number of dead and deformed ones by the total number of pupae. Once adults emerged from those pupae under a different storage period, they were coupled randomly (one female and one male) and placed into the oviposition cage (plastic transparent cuboid cage, 6.5 × 6.5 × 9.5 cm) supplied with 10% sugar solution. Longevity and fecundity of these female adults were investigated according to the method described above.

### 2.4. Oviposition Pattern and Larval Mortality

To test preferences of oviposition sites of *P. ringoniella* females, 18 three-month-old crabapple seedlings (nearly 20 cm height) with five mature leaves were placed in the breeding cage for 24 h. The breeding cage contained 5–10 female adults. The leaves on each plant were marked as 1, 2, 3, 4, and 5 from low to high, respectively (Figure 1A). The eggs laid on each marked leaf were counted using a binocular stereomicroscope. The left and right sides of stereomicroscope were lifted by two wooden blocks (16 × 10 × 12 cm), so that there was some space (about 12 cm width) under the stereomicroscope and the crabapple seedling (Figure 1B) could be placed through the hole in the bottom of microscope. The eggs on each leaf were observed and counted carefully without harming the seedling. The eggs were laid mostly near the leaf vein, thus we separated the eggs laid on each leaf into four classes according to their nearness to the respective leaf vein: midrib, secondary, tertiary, and margin (Figure 1B). The proportions of the eggs within these four classes were calculated.

A total of 70 crabapple seedlings containing 280 leaves were investigated after being exposed to *P. ringoniella* in the breeding cage (40 × 40 × 40 cm) for 24 h. The breeding cage was maintained at 25 ± 1 °C, 65 ± 5% RH, and a photoperiod of 14:10 (L:D) h and contained 5–10 female adults. The number of eggs laid on each leaf was counted using a binocular stereomicroscope.

Because of the cannibalism among the *P. ringoniella* larvae [8], our hypothesis was that the larval mortality increased with the number of larvae per leaf. To test this hypothesis, larval mortality on each leaf (with 1–6 larvae) was investigated in the growth chamber at 25 ± 1 °C, 65 ± 5% RH, and a photoperiod of 14:10 (L:D) h. This experiment included four replications and each replication included 50 leaves. The initial numbers of larvae on each leaf were checked and then larvae survived into pupae were counted to estimate the mortality during larval period. We computed average larval mortality for each initial larval density.

### 2.5. Data Analysis

Statistical analyses were conducted using SAS 9.3 [16]. The proportion of eggs laid on different positions was compared by one-way ANOVA, and the means were compared by Tukey’s honest significant difference (HSD) test. The relationship between larval mortality and the number of larvae per leaf was analyzed by linear regression procedure. The longevity, preoviposition, oviposition and postoviposition periods, and total fecundity were evaluated using one-way ANOVA, and the means were separated by Tukey’s HSD test. The average longevities of females and males at each temperature were compared with paired *t*-test. The developmental period, mortality, and deformation rate of pupae after different storage periods were compared by one-way ANOVA, and means were separated by Tukey’s HSD test. Sex ratio was analyzed using Chi-square test. The body sizes of females and males were comparted by *t*-test. The data of proportion, mortality, and deformation rate were transformed by arcsine square root to normalize the variance before analyses.

## 3. Results

### 3.1. Rearing P. ringoniella Using Crabapple Leaves and Morphological Measurement

According to our rearing method described above, the mean developmental period was 25.9 ± 0.49 days from egg to adult, and the survival rate was 0.387 (Table 1). The length of the larval developmental period was 59.6% of the total immature period, which was higher than that of eggs (18.5%) and pupae (21.9%). The larval survival rate (0.504) was lower than that of eggs (0.873) and pupae (0.879). The sex ratio (female proportion) was 0.48 (Table 1). Provision with 10% sugar solution resulted in mean longevity of ovipositing females of 6.1 ± 0.33 days and the fecundity (mean egg number laid per female) was 42.6 ± 5.30 (Table 2). This rearing method had been carried out over five generations without problems.

After hatching, the larvae burrow directly into the leaf and extend its mine by shearing through spongy mesophyll cells adjacent to the lower epidermal cells. During the sap feeder stage, the mine appears as a silvery-green spot of irregular shape on the lower surface of the leaf. The sap feeders are legless and have a flattened wide head. The body is white and semitransparent, with a wider thorax and a narrow and shorter abdomen. As the larvae developed into tissue feeders, they began to chew out palisade cells and epidermis of the upper surface of the leaves, which resulted in a greenish-white appearance on the upper surface of the leaves. The tissue feeders have three pairs of true legs, three pairs of prolegs in third, fourth, and fifth segments of abdomen, and one pair of anal prolegs in the ninth segment of the abdomen.

Body sizes of the different stages of *P. ringoniella* at 25 °C are presented in Table 3. Average lengths and widths of the eggs were 0.336 ± 0.0043 and 0.259 ± 0.0046 mm, respectively. Body length increased from 1.070 ± 0.0245 in first instar larvae to 5.027 ± 0.0718 mm in fifth instar larvae. The head width of the larvae ranged from 0.180 ± 0.0021 to 0.321 ± 0.0021 mm. Compared with fifth instar larva, the body length of pre-pupa decreased to 4.705 ± 0.0520 mm (Table 3). The body length of the pupae (3.643 ± 0.0414 mm) was shorter than that of the larvae and pre-pupae, while the head region was wider than in the larvae and pre-pupae. In addition, the weight of the pupae was 0.946 ± 0.0132 mg. There were no significant differences in body length (*t*-test, t = 1.89, df = 58, *p* = 0.0636) and antennal lengths (t = 0.92, df = 58, *p* = 0.3634) between females and males. The difference between the length of the body and the antenna in females was not significant (t = 0.65, df = 58, *p* = 0.5181), while it was significant in males (t = 2.35, df = 58, *p* = 0.0224). However, the wingspan of males (6.280 ± 0.0639 mm) was significantly larger than that of females (6.040 ± 0.0753 mm) (t = 2.43, df = 58, *p* = 0.0182).

### 3.2. Effects of Diets on Adult Performance

At a temperature of 25 °C, the diet significantly influenced the longevity of the females (ANOVA, Tukey’s test, F = 7.26, df = 3,104, *p* = 0.0002) and males (F = 12.62, df = 3,104, *p* < 0.0001) (Table 2). Both females and males can survive longer (8.2 ± 0.55 and 9.4 ± 0.55 days, respectively) when supplied with 10% honey solution. Surprisingly, the adults can survive more than five days even in the control (with no food and water) (Table 2). The survival curves and oviposition rates of *P. ringoniella* females supplied with different diets are shown in Figure 2. The fecundity (total number of eggs per female) was significantly (F = 7.46, df = 3,104, *p* = 0.0001) higher when insects were supplied with 10% honey compared with remaining adult diets. The oviposition period, female and male longevity of adults on honey diet were also significantly better, whereas the preoviposition and postoviposition periods were not (Table 2).

### 3.3. Effect of Low Temperature Storage of Pupae on Adult Performance

The pupae were stored at 4 °C with increases from 0 days to 105 days separated by 15 days interval. The developmental periods were significantly different after different storage periods at 4 °C (Table 4; F = 1.32, df = 7,420, *p* < 0.0001). The developmental periods ranged from 4.2 ± 0.14 to 6.1 ± 0.08 days. Pupal mortality and deformation rates increased significantly as the storage period became longer (Table 4). The mortality of the pupae increased from 5.6 ± 4.01 to 51.1 ± 9.88%, as the storage periods increased from 0 to 105 days. The deformation rate ranged from 4.4 ± 2.22 to 30.0 ± 5.09% (Table 4). After different storage periods at 4 °C, sex ratios (female proportion of total adults) of the emerged adults did not change and were 0.5 (Table 4; χ^2^ = 5.6157, df = 7, *p* = 0.5853).

The reproductive characteristics of adults that emerged from the pupae after different storage periods at 4 °C are presented in Table 5. Longevity and fecundity of female adults were not influenced by the storage period; the same goes for male longevity. Preoviposition, oviposition, and postoviposition periods were also not significantly affected by storage period. However, the storage period of 45 days showed the maximum reproductive characteristics of adults mentioned above (Table 5).

### 3.4. Oviposition Pattern and Larval Mortality

There was no significant preference of eggs laid on the different leaves of one plant (Figure 1A; F = 2.34, df = 4,85, *p* = 0.0619). However, female adults preferred to lay eggs (56.5 ± 3.84%) near the secondary leaf vein of a leaf (Figure 1B; F = 50.46, df = 3,65, *p* < 0.0001).

Eggs were laid singly, and mostly one to two eggs per leaf. The frequency distribution of eggs laid per leaf exhibited the positive skewed distribution pattern shown in Figure 3.

Larval mortality increased linearly (r^2^ = 0.94) with the number of larvae per leaf (Figure 4). The larval mortality was lower (7.8 ± 1.22%) as one larva developed per leaf, while it increased (61.1 ± 20.03%) when six larvae developed together in one leaf.

## 4. Discussion

This study provided a rearing method for *P. ringoniella* in the laboratory. Jung and Boo (1977) [17] developed an artificial diet for rearing *P. ringoniella* larvae, and reported the problem of diet contamination with microorganisms and the difficulty of supplying artificial diet to first instar larvae. However, we used the crabapple seedlings for egg oviposition and larval feeding. The crabapple seedlings do not need a wide space or equipment for planting and growing in the laboratory, but can be provided with the same quality throughout the year. The method described in this study can be used for mass rearing and colony maintenance for *P. ringoniella* in the laboratory. For mass rearing, it is crucial to develop a system requiring low preparation costs and labor, and to obtain a maximum number of individuals during a short period. Moreover, handling times for carrying out the rearing procedures, such as egg laying, pupae collection, and providing foods for adults took less than five minutes per breeding cage per day. In addition, the pupae could be stored at low temperature for more than three months. Thus, the method is eminently convenient for colony maintenance and discontinuous rearing of *P. ringoniella* in the laboratory.

In the present study, we observed and measured the morphological characteristics of different stages of *P. ringoniella*. The eggs are laid on the lower surface of the leaves. Newly laid eggs are faint yellow and orbicular with lustrous colors on the surface. The lustrous color will fade away as the egg developed. Lengths and widths were measured as 0.336 and 0.259 mm, respectively. The larvae had been classified into five instars [18,19], while some researcher classified the larvae into sap feeder and tissue feeder for convenience in the field surveys [8]. Sap feeders are the younger larvae that feed on the spongy mesophyll cells, while tissue feeders are the older ones which feed on the palisade cells. According to our observations, the first and second instar larvae are sap feeders and the third to fifth instar larvae are tissue feeders. The biology of the larvae in our colony is similar to that described by Sekita and Yamada (1979) [8]. The head widths of first to fourth instar larvae measured in this study were similar to those reported by Wang et al. (2007) [18]. The difference occurred in the fifth instar larvae, which were measured as 0.321 ± 0.0021 mm in the present study, while Wang et al. (2007) [18] determined them as 0.3778 mm. This difference may be caused by the sample resource: Wang et al. (2007) [18] collected larvae from the apple orchard in China, while we sampled from the laboratory colony reared on crabapple in Korea. The body length became shorter when fifth instar larvae changed into the pupal stage. Pupae are initially yellow in color, but they darken into brown coloration over time. Adults emerge through the surface of the leaf, leaving the pupal case partially sticking out of the empty mine. The results showed that the males had larger wingspans than the females. The antennal length of the male was longer than its body, while there was no difference between body and antennal lengths in the females.

The total fecundity (47.6 ± 7.03) per female provided with 10% sugar solution in this study, was similar to that reported in a previous study which gave a figure of approximate 40 eggs per female feeding on 10% sugar solution [4,20]. Compared with the control, longevity and fecundity of *P. ringoniella* females improved when they were supplied with water, 10% sugar, and 10% honey solutions. Moreover, when supplied honey solution, oviposition period and longevity increased. Even though there was no statistical difference, fecundity increased 120% of sugar solution-fed females. This result implies that adult females may benefit from feeding on nectar or honey, which are good sources of energy, minerals, and vitamins [21].

The frequency distribution of eggs laid per leaf can directly reflect the pattern of adult oviposition. The number of eggs laid on the five different leaves in one seeding was not significantly different (Figure 1A). A possible reason was that the physiological status of the five leaves was not so much different. In the field condition, *P. ringoniella* infested different leaf sites with different generations [2]. Li et al. (2017) [13] also found the significant migration of *P. ringoniella* from lower to upper parts of the apple tree, and speculated the adult moth would prefer the fresh leaves, which are distributed in the higher position of the tree in late summer. However, in our study, the five leaves were newly emerged from the seedling at about the same time and offered little variation to base preference on.

The eggs are laid singly on the lower surface of the leaf, and usually alongside leaf veins including midrib, secondary vein, and tertiary vein, and seldom on the marginal area of a leaf. The leaf veins can determine the fixed form of the mine cut out by sap feeding larvae. The sap feeding larvae are able to cleave small leaf veins (tertiary), but they cannot cleave the larger ones (midrib and secondary). When they meet larger veins, rather than cleave them, they change their course along the vein until they meet another large vein. Through such repeated processes, the more or less definite form of mine is constructed. This oviposition behavior has its advantages: reducing the exposure probability to predators and avoiding the cannibalism occurring during larval development. The eggs laid near the leaf veins are difficult to discover by predators. Sekita and Yamada (1979) [8] reported the occurrence of cannibalism needs two basic conditions: (1) the eggs are laid so close to each other that the resulting mines are situated within the span of overlapping; and (2) there is no obstructive vein between the eggs. In this study, most eggs were laid near the secondary vein, which can serve as obstructive vein for the limited area of the mine and further to avoid the occurrence of cannibalism.

Although we did not quantitatively classify the mortality factor during larval stages, we observed that the larval mortality was largely due to cannibalism. Sekita and Yamada (1979) [8] reported that cannibalism was the only density-dependent factor of larval mortality among factors such as parasites, cannibalism, and unknown factors. We also found that the mortality of larvae was dependent on larval density (number of larvae per leaf) in this study. This finding is consistent with the observation that extremely high adult populations during the year do not result in as high level of damage rates on leaves as would be predicted, partly due to cannibalism [2,13].

The pupae could be stored at a cold temperature 4 °C for 105 days, even the mortality had reached 51.1%. Since the late larval and pupal stage experience cold winter, but not the eggs and larvae, storing for long term for the research and other purpose such as breeding biocontrol agents would benefit to the pupal stage. Cold storage of pupae can facilitate discontinuous rearing and enhance the mass production of *P. ringoniella* in the laboratory. After cold storage, some pupae emerged to be aberrant adults, which failed to expand their wings. The deformation rate increased as the storage periods became longer. Similar result was reported from Mattoni et al. (2003) [22] showing higher defective wing patterns of *Glaucopsyche lygdamus palosverdesensis*, an endangered butterfly, while they were mass reared for restoration. Detailed mechanisms could be explored further to overcome challenges.

## 5. Conclusions

Overall, we developed a simple rearing method for *P. ringoniella* and measured its morphological characteristics in the laboratory, tested diets and low temperature storage of pupae on adult performance, and described its oviposition pattern. This study will be a useful contribution in connection with further studies on this pest insect, such as demographic determination, temperature-dependent development, oviposition model development, pesticide toxicity testing, and population dynamic model construction.

## Figures and Tables

**Figure 1 insects-10-00387-f001:**
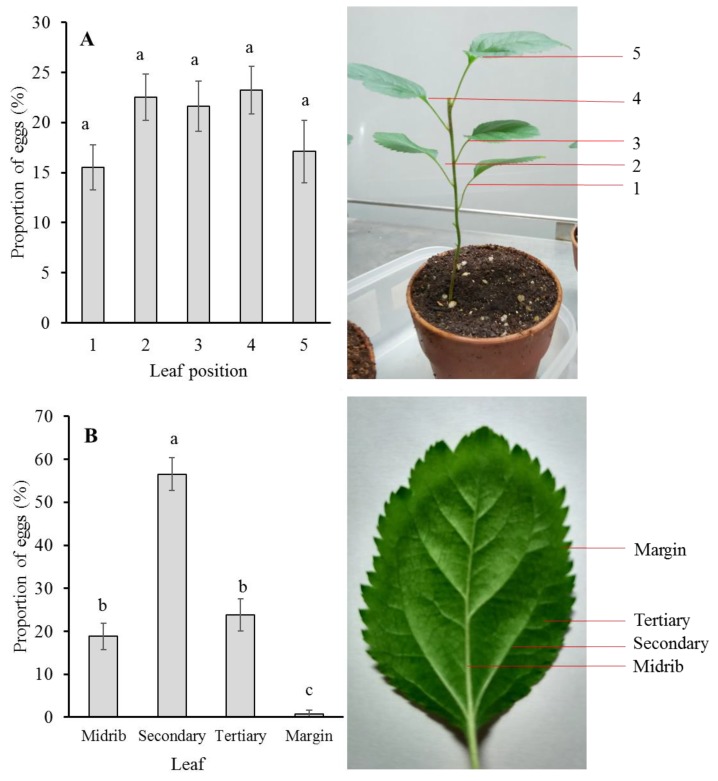
The proportion of eggs (±SE) laid by *Phyllonorycter ringoniella* adult females on different leaves within one crabapple seedling plant (**A**) and at different leaf veins within one leaf (**B**). Means labeled with same letters are not significantly different (Tukey HSD test; A: F = 2.34, df = 4,85, *p* = 0.0619; B: F = 50.46, df = 3,65, *p* < 0.0001).

**Figure 2 insects-10-00387-f002:**
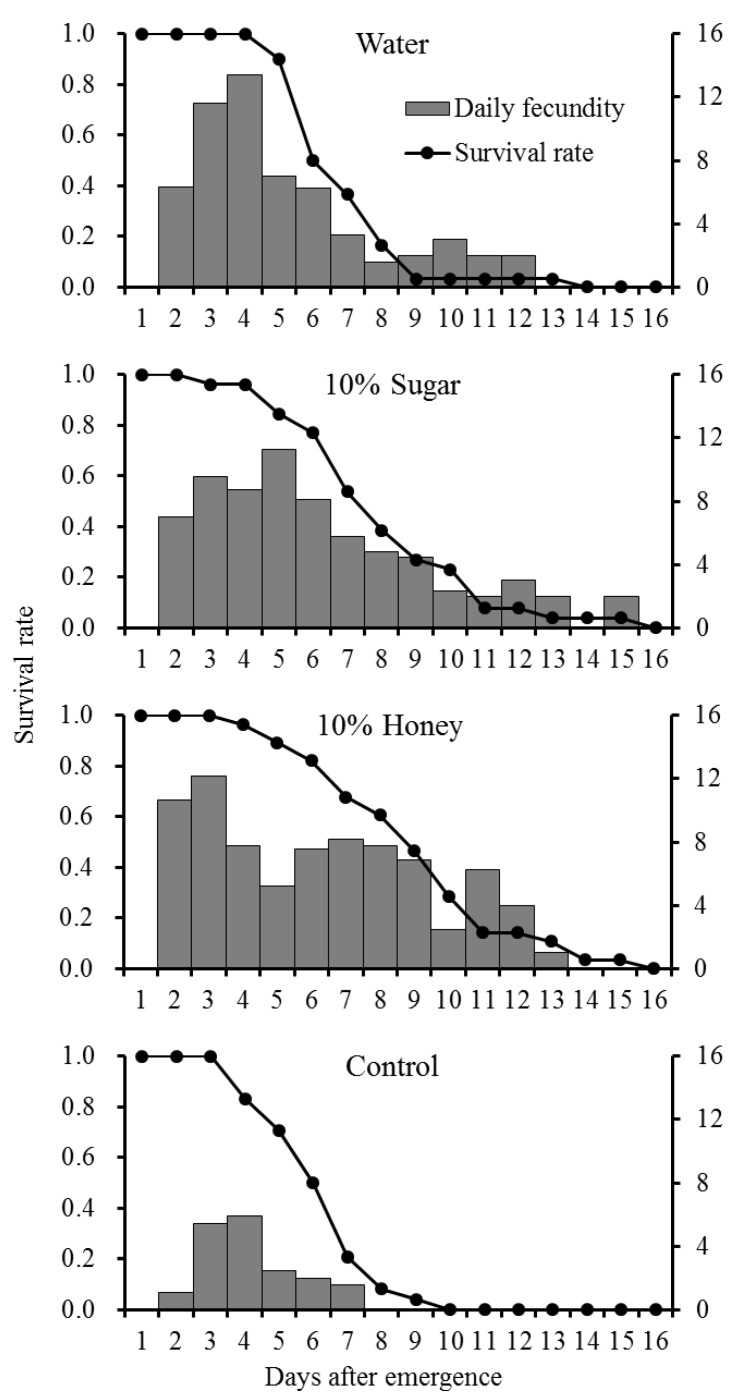
Survival curve and daily fecundity of *Phyllonorycter ringoniella* females supplied with water, 10% sugar, 10% honey, and nothing (control) at 25 °C.

**Figure 3 insects-10-00387-f003:**
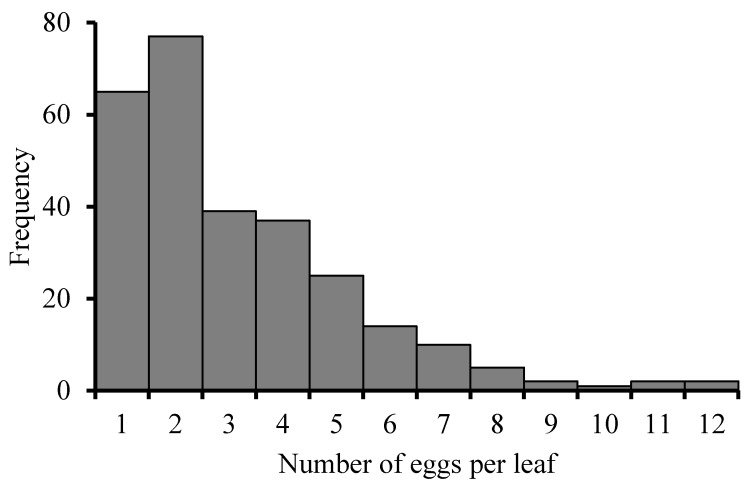
Frequency distribution of the number of *Phyllonorycter ringoniella* eggs per leaf.

**Figure 4 insects-10-00387-f004:**
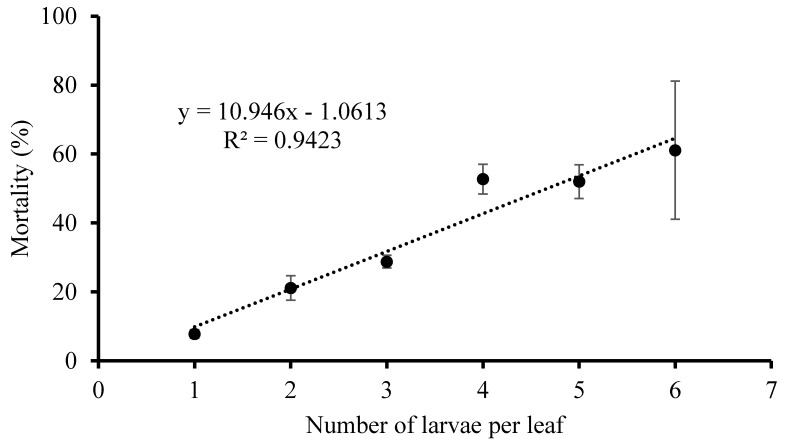
Relationship between initial larval density and mortality (±SE) during larval period of *Phyllonorycter ringoniella* on the crabapple leaf in the laboratory.

**Table 1 insects-10-00387-t001:** Mean developmental periods (±SE), survival rate, and sex ratio of *Phyllonorycter ringoniella* immature stages under 25 °C.

Stage	N *	Developmental Period (days)	Survival Rate	Sex Ratio **
Egg	150	4.8 ± 0.06	0.873	-
Larva	131	15.5 ± 0.39	0.504	-
Pupa	66	5.7 ± 0.07	0.879	-
Total immature	58	25.9 ± 0.49	0.387	0.48

* Initial number of individuals at each stage. ** Female proportion = Number of female adults/number of total adults.

**Table 2 insects-10-00387-t002:** Effect of diet provision on the adults’ preoviposition, oviposition, postoviposition, and longevity (days), and fecundity (mean ± SE) of *Phyllonorycter ringoniella* under 25 °C.

Diet	N *	Preoviposition Period	Oviposition Period	Postoviposition Period	Longevity		Fecundity	Daily Fecundity
					Female	Male		
Water	30	2.0 ± 0.23 a	3.3 ± 0.33 bc	0.8 ± 0.15 a	6.1 ± 0.33 bc	7.1 ± 0.40 b	42.6 ± 5.30 a	7.2 ± 0.84 a
10% Sugar	26	1.7 ± 0.21 a	4.0 ± 0.57 ab	1.6 ± 0.38 a	7.2 ± 0.55 ab	8.7 ± 0.76 ab	47.6 ± 7.03 a	6.8 ± 0.81 a
10% Honey	28	1.9 ± 0.27 a	4.9 ± 0.57 a	1.4 ± 0.19 a	8.2 ± 0.55 a	9.4 ± 0.55 a	56.9 ± 8.58 a	7.0 ± 0.98 a
No diet	24	2.3 ± 0.22 a	2.2 ± 0.25 c	0.9 ± 0.21 a	5.4 ± 0.33 c	5.0 ± 0.23 c	14.5 ± 2.90 b	2.9 ± 0.59 b

* Number of adult individuals. Means followed by the different letters in the column are significantly different (ANOVA, *p* < 0.05, Tukey’s HSD test; Preoviposition: F = 1.32, *p* = 0.0705; Oviposition: F = 6.26, *p* = 0.0006; Postoviposition: F = 2.38, *p* = 0.0735; Female longevity: F = 7.26, *p* = 0.0002; Male longevity: F = 12.62, *p* < 0.0001; Fecundity: F = 7.46, *p* = 0.0001; Daily fecundity: F = 5.51, *p* = 0.0015; df = 3,104 in all cases).

**Table 3 insects-10-00387-t003:** Body sizes (mean ± SE) of different stages of *Phyllonorycter ringoniella*.

Stage	N *	Length (mm)	Width (mm)	
Egg	30	0.336 ± 0.0043	0.259 ± 0.0046	
		Body length (mm)	Head width (mm)	Weight (mg)
Larva				
1st instar	30	1.070 ± 0.0245	0.180 ± 0.0021	-
2nd instar	30	2.022 ± 0.0467	0.238 ± 0.0022	-
3rd instar	49	3.124 ± 0.0679	0.261 ± 0.0049	-
4th instar	35	4.155 ± 0.0799	0.306 ± 0.0039	-
5th instar	33	5.027 ± 0.0718	0.321 ± 0.0021	-
Pre-pupa	40	4.705 ± 0.0520	0.322 ± 0.0022	-
Pupa	56	3.643 ± 0.0414	0.374 ± 0.0034	0.946 ± 0.0132
Adult		Body length (mm)	Antenna (mm)	Wingspan (mm)
Female	30	2.493 ± 0.0377	2.460 ± 0.0347	6.040 ± 0.0753
Male	30	2.400 ± 0.0318	2.502 ± 0.0294	6.280 ± 0.0639

* Number of tested individuals.

**Table 4 insects-10-00387-t004:** Developmental periods, mortality, deformation rate, and sex ratio (mean ± SE) of *Phyllonorycter ringoniella* pupae under 25 °C after different storage periods of early pupal stage at 4 °C.

Storage Period (Days)	N *	Developmental Period (Days)	Mortality (%)	Deformation Rate (%)	Sex Ratio **
0	90	5.8 ± 0.07 ab	5.6 ± 4.01 c	4.4 ± 2.22 b	0.50 ± 0.041
15	90	6.1 ± 0.08 a	14.4 ± 1.11 bc	6.7 ± 1.92 ab	0.49 ± 0.017
30	90	4.7 ± 0.11 cd	10.0 ± 0.00 c	5.6 ± 1.11 ab	0.53 ± 0.118
45	90	4.5 ± 0.12 de	16.7 ± 0.00 bc	12.2 ± 4.01 ab	0.47 ± 0.013
60	90	5.5 ± 0.12 b	18.9 ± 6.19 bc	28.9 ± 8.68 a	0.54 ± 0.080
75	90	4.2 ± 0.14 e	35.6 ± 7.78 ab	27.8 ± 8.68 a	0.46 ± 0.095
90	90	5.0 ± 0.25 c	42.2 ± 2.22 a	30.0 ± 5.09 a	0.63 ± 0.048
105	90	4.9 ± 0.13 cd	51.1 ± 9.88 a	14.4 ± 4.84 ab	0.47 ± 0.042

* Number of tested pupae. ** Female proportion = Number of female adults/number of total adults, sex ratio was 0.5 and was unaffected by the storage periods (Chi-square test, χ^2^ = 5.6157, df = 7, *p* = 0.5853). Means followed by the different letters in the column are significantly different (ANOVA, *p* < 0.05, Tukey’s HSD test; Developmental period: F = 1.32, df = 7,420, *p* < 0.0001; Mortality: F = 9.16, df = 7,16, *p* = 0.0001; Deformation rate: F = 4.86, df = 7,16, *p* = 0.0042).

**Table 5 insects-10-00387-t005:** Preoviposition, oviposition, postoviposition, longevity (days), and fecundity (mean ± SE) of *Phyllonorycter ringoniella* developed from different storage periods of pupae at 4 °C.

Storage Period (Days)	N *	Preoviposition	Oviposition	Postoviposition	Longevity		Fecundity	Daily Fecundity
					Female	Male		
0	26	1.7 ± 0.21 a	4.0 ± 0.57 a	1.6 ± 0.38 a	7.2 ± 0.55 a	8.7 ± 0.76 a	47.6 ± 7.03 a	6.8 ± 0.81 a
15	24	1.8 ± 0.23 a	5.0 ± 0.52 a	1.1 ± 0.27 a	7.9 ± 0.57 a	9.0 ± 0.61 a	63.5 ± 8.91 a	8.0 ± 0.99 a
30	19	1.6 ± 0.19 a	3.6 ± 0.57 a	1.8 ± 0.41 a	6.9 ± 0.64 a	6.9 ± 0.67 a	35.2 ± 9.23 a	4.8 ± 1.13 a
45	25	2.3 ± 0.33 a	4.7 ± 0.60 a	2.1 ± 0.55 a	9.1 ± 0.60 a	7.4 ± 0.59 a	44.0 ± 8.35 a	4.9 ± 0.83 a
60	12	2.2 ± 0.41 a	2.6 ± 0.50 a	1.3 ± 0.57 a	6.1 ± 0.70 a	6.3 ± 0.95 a	23.2 ± 7.41 a	4.0 ± 1.10 a
75	8	2.4 ± 0.53 a	4.9 ± 1.57 a	1.1 ± 0.35 a	8.4 ± 1.65 a	9.6 ± 0.89 a	34.6 ± 17.66 a	3.9 ± 1.30 a
90	9	1.9 ± 0.54 a	5.4 ± 1.06 a	1.6 ± 0.38 a	8.9 ± 1.16 a	8.2 ± 0.98 a	46.0 ± 10.31 a	5.4 ± 1.10 a
105	9	1.7 ± 0.37 a	4.4 ± 1.07 a	1.7 ± 0.44 a	7.8 ± 0.94 a	9.1 ± 1.11 a	35.0 ± 10.74 a	4.1 ± 1.26 a

* Number of adult individuals. Means followed by the same letters in the column are not significantly different (ANOVA, *p* > 0.05, Tukey’s HSD test; Preoviposition: F = 0.92, *p* = 0.4949; Oviposition: F = 1.36, *p* = 0.2277; Postoviposition: F = 0.61, *p* = 0.7472; Female longevity: F = 1.79, *p* = 0.0955; Male longevity: F = 1.86, *p* = 0.0812; Fecundity: F = 1.67, *p* = 0.1234; Daily fecundity: F = 2.15, *p* = 0.0436; df = 7124 in all cases).

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
