# Peer review of "Effect of Diets and Low Temperature Storage on Adult Performance and Immature Development of Phyllonorycter ringoniella in Laboratory"

_insects, 2019, doi:10.3390/insects10110387_

Round 1

Reviewer 1 Report

Review report of manuscript ID: insects-604574] Entitled: Effect of diets and low temperature storage on adult performance and immature development of Phyllonorycter ringoniella in Laboratory

Line no          Original (1)                                                  Change proposed (2)

78   1    The mean developmental periods and survival rates of eggs, larvae, pupae, and total immature stages were calculated

2   How the survival rates of eggs, larvae, pupae, and total immature stages were calculated?

98 1 A piece of cotton moistened with different diet was placed in each cage as food for the adults

2   A piece of cotton moistened with different diets was placed in each cage as food for the adults

112 1   thirty pupae individuals

2   thirty individual pupae

118   1   they were coupled (one female and one male) and placed into the oviposition cage

2   they were coupled randomly? (one female and one male) and placed into the oviposition cage

156   1   the mean developmental period was 25.9 days from egg to adult, and the survival rate was 0.387 (Table 1).

2   the mean developmental period was 25.9 ± 0.49 from egg to adult, and the survival rate was 0.387 (Table 1).Please give (± SE) whenever describing any development stage in results.

316   1    can served

2     can serve

191    1      The fecundity (total number of eggs per female) was lowest  in the control, while there were no differences of fecundities when they were supplied with water, 192 10% sugar, and 10% honey (F = 7.46, df = 3, 104, P = 0.0001). The oviposition period was significantly 193 affected by diet, whereas the preoviposition and postoviposition periods were not (Table 2).

2     The fecundity (total number of eggs per female) was significantly (F = 7.46,  df = 3, 1043,P = 0.0001) higher when insects were supplied with 10% honey compared with remaining adult diets. The oviposition period, female and male longevity of adults on honey diet were also significantly better, whereas the preoviposition and postoviposition periods were not (Table 2).

205    1      Longevity and fecundity of female adults were not influenced 205 by the storage period; the same goes for male longevity. Preoviposition, oviposition, and 206 postoviposition periods were also not affected by storage period (Table 5).

2     Longevity and fecundity of female adults were not influenced by the storage period; the same goes for male longevity. Preoviposition, oviposition, and postoviposition periods were also not significantly affected by storage period. However, the storage period of 45 days showed the maximum reproductive characteristics of adults mentioned above (Table 5).

Author Response

Thank you very much for giving these comments and corrections on our manuscript. Here are our responses (written in red) to reviewers.

Reviewer 1

Line no          Original (1)                                                  Change proposed (2)

78   1    The mean developmental periods and survival rates of eggs, larvae, pupae, and total immature stages were calculated

2   How the survival rates of eggs, larvae, pupae, and total immature stages were calculated?

Author: I had added the calculated methods in the revised manuscript.

98 1 A piece of cotton moistened with different diet was placed in each cage as food for the adults

2   A piece of cotton moistened with different diets was placed in each cage as food for the adults

Author: I had accepted your suggestion in the revised manuscript.

112 1   thirty pupae individuals

2   thirty individual pupae

Author: I had accepted your suggestion in the revised manuscript.

118   1   they were coupled (one female and one male) and placed into the oviposition cage

2   they were coupled randomly? (one female and one male) and placed into the oviposition cage

Author: I had accepted your suggestion in the revised manuscript.

156   1   the mean developmental period was 25.9 days from egg to adult, and the survival rate was 0.387 (Table 1).

2   the mean developmental period was 25.9 ± 0.49 from egg to adult, and the survival rate was 0.387 (Table 1).Please give (± SE) whenever describing any development stage in results.

Author: I had accepted your suggestion in the revised manuscript.

316   1    can served

2     can serve

Author: I had accepted your suggestion in the revised manuscript.

191    1      The fecundity (total number of eggs per female) was lowest  in the control, while there were no differences of fecundities when they were supplied with water, 192 10% sugar, and 10% honey (F = 7.46, df = 3, 104, P = 0.0001). The oviposition period was significantly 193 affected by diet, whereas the preoviposition and postoviposition periods were not (Table 2).

2     The fecundity (total number of eggs per female) was significantly (F = 7.46,  df = 3, 1043,P = 0.0001) higher when insects were supplied with 10% honey compared with remaining adult diets. The oviposition period, female and male longevity of adults on honey diet were also significantly better, whereas the preoviposition and postoviposition periods were not (Table 2).

Author: I had accepted your suggestion in the revised manuscript.

205    1      Longevity and fecundity of female adults were not influenced 205 by the storage period; the same goes for male longevity. Preoviposition, oviposition, and 206 postoviposition periods were also not affected by storage period (Table 5).

2     Longevity and fecundity of female adults were not influenced by the storage period; the same goes for male longevity. Preoviposition, oviposition, and postoviposition periods were also not significantly affected by storage period. However, the storage period of 45 days showed the maximum reproductive characteristics of adults mentioned above (Table 5).

Author: I had accepted your suggestion in the revised manuscript.

Reviewer 2 Report

This article, “effect of diet and low-temperature storage…..Phyllonorycter ringoniella in the laboratory” by Gen et al. reports information regarding rearing conditions for an economic pest of fruits, mainly apple in several Asian counties. The primary goal of this study was to find out economic rearing condition of this leaf miner pest in the laboratory. Authors have used crabapples as a rearing host and tried few diets for adult feeding to evaluate the effect of these diets on oviposition and other life-history traits. Another aspect of the study was to evaluate the length of time the pupa of this insect can be stored without compromising the viability of adults. Besides these two main aspects, authors have also documented that where in the leaf of crabapple this insect prefers to oviposit and if there is any larval mortality due to cannibalistic effect. Overall, this information is important for establishing a rearing facility for any insect pest that is used for research purposes.

The research presented in this article was well written and easy to follow. I did not find any significant flaws with the usage of the English language except a few editorial or unintentional mistakes. It seems that the research was conducted with planned experimental setup and data was collected to answer specific questions the authors had in mind. However, I have a few comments/suggestions that might help to improve the manuscript. Please see the following:

I believe that the diet of larvae highly influences the performance of adults. There is plenty of literature on this topic. Therefore, when your title says “effect of diets”, it is likely that readers would perceive this study differently. I suggest changing your title to reflect the study. Is there any other possible host plant, which can be used instead of crabapple? Possibly, there are better host plants for this species and thus can help in better rearing outcomes. List any references on this matter. I was curious how authors were able to manipulate live plants under the microscope for taking various counts. It can be helpful to other readers as well if authors can add more details on the method section (line 127). In the abstract section, along with mean use SE. Being consistent would be good. The outbreak of “this” pest (Line 42) Delete “an” (since there were more than one application) of pesticides (line 42) I am not sure if the use of “morphometric” is right. I believe it is related to the shape aspect of morphological characters. So, why not use “morphological” instead of “morphometric”? The sex ratio was determined “from “ the emerged adults ( delete “by” in line 80) Rephrase this sentence (line 131-135). Leaves were kept inside the breeding cage, not exposed to the breeding cage. You exposed the plants to insects. In the result section, paragraph (line 164-172) is not necessary. It can be moved to discussion if you feel important. “wingspan”, not plural (line 183) Delete “responsible from” and add “due to” the cannibalism (line 319) Not sure if you wanted to mean “discontinuous” or “continuous” (line 266). Please check.

Author Response

Reviewer 2

This article, “effect of diet and low-temperature storage…..Phyllonorycter ringoniella in the laboratory” by Gen et al. reports information regarding rearing conditions for an economic pest of fruits, mainly apple in several Asian counties. The primary goal of this study was to find out economic rearing condition of this leaf miner pest in the laboratory. Authors have used crabapples as a rearing host and tried few diets for adult feeding to evaluate the effect of these diets on oviposition and other life-history traits. Another aspect of the study was to evaluate the length of time the pupa of this insect can be stored without compromising the viability of adults. Besides these two main aspects, authors have also documented that where in the leaf of crabapple this insect prefers to oviposit and if there is any larval mortality due to cannibalistic effect. Overall, this information is important for establishing a rearing facility for any insect pest that is used for research purposes.

The research presented in this article was well written and easy to follow. I did not find any significant flaws with the usage of the English language except a few editorial or unintentional mistakes. It seems that the research was conducted with planned experimental setup and data was collected to answer specific questions the authors had in mind. However, I have a few comments/suggestions that might help to improve the manuscript. Please see the following:

I believe that the diet of larvae highly influences the performance of adults. There is plenty of literature on this topic. Therefore, when your title says “effect of diets”, it is likely that readers would perceive this study differently. I suggest changing your title to reflect the study.

Author: Thank you very much for your kind suggestion on our title, we also had discussed more on the title, previous titles such as “The laboratory rearing method and biological characteristics of Phyllonorycter ringoniella”, “The Rearing Method and Biological Characteristics of Phyllonorycter ringoniella in Laboratory”, “Effect on the food source on adult performance and immature development of Phyllonorycter ringoniella”. As you said, this study contained rearing method, oviposition trait, diets on adults, biological characteristic, pupae storage, and larval mortality. As a result, we had chosen “Effect of diets and low temperature storage on adult performance and immature development of Phyllonorycter ringoniella in Laboratory” as the title, because it can reflect the major and key points of this study.

Is there any other possible host plant, which can be used instead of crabapple? Possibly, there are better host plants for this species and thus can help in better rearing outcomes. List any references on this matter.

Author: The hosts of P. ringnoniella include some pome and stone fruit trees such as apple, pear, peach, cherry, and plum (Kumata et al. 1983, Du et al. 2013). It had been reported rearing this leafminer successfully with crabapple (Sun et al. 2007), and the crabapple seedlings were easy to plant in laboratory. Thus, we selected crabapple as host plant for this species, the related references were also followed in the manuscript.

I was curious how authors were able to manipulate live plants under the microscope for taking various counts. It can be helpful to other readers as well if authors can add more details on the method section (line 127).

Author: We used Olympus SZ51 stereomicroscope, first we moved the species man stage from the base, then put two wooden blocks (12 cm height) under two sides of the stereomicroscope, thus, we can put the crabapple seedling through the hole and count the eggs on each leaf carefully by adjustment arm.

We made some detailed information in the revised manuscript.

In the abstract section, along with mean use SE. Being consistent would be good.

Author: Thank you for your suggestion, we keep them consistent by mean ± SE.

The outbreak of “this” pest (Line 42) Delete “an” (since there were more than one application) of pesticides (line 42)

Author: We delete “an”.

I am not sure if the use of “morphometric” is right. I believe it is related to the shape aspect of morphological characters. So, why not use “morphological” instead of “morphometric”?

Author: Thank you for your suggestion, we used “morphological” instead of “morphometric”.

The sex ratio was determined “from “ the emerged adults ( delete “by” in line 80)

Author: We changed “by” to “from”

Rephrase this sentence (line 131-135). Leaves were kept inside the breeding cage, not exposed to the breeding cage. You exposed the plants to insects.

Author: We changed in the revised manuscript.

In the result section, paragraph (line 164-172) is not necessary. It can be moved to discussion if you feel important.

Author: We think that this paragraph reflect our observation result, we prefer to keep it in result section.

“wingspan”, not plural (line 183)

Author: Thank you, we changed to “wingspan”.

Delete “responsible from” and add “due to” the cannibalism (line 319)

Author: Thank you, we changed to “responsible from” to “due to”.

Not sure if you wanted to mean “discontinuous” or “continuous” (line 266). Please check.

Author: I mean discontinuous rearing.

Reviewer 3 Report

The manuscript by Geng et al. is well written and provides valuable information about laboratory methods of rearing the Asiatic apple leaf miner. Except for a few comments (see attached pdf with comments for details), I think this manuscript should be accepted for publication after those comments have been addressed.

Author Response

Reviewer 3

It is confusing because the first result presented in that about the egg dimensions, whereas, the authors only mention the collection of overwintering pupae earlier. This seems abrupt.

Author: We changed the sentence sequence in the revised manuscript.

It may be more helpful, if % increase or decrease is provided here and in results section, rather than means

Author: We prefer to use means to present their characteristic.